# Genetically Encoded Biosensors to Monitor Intracellular Reactive Oxygen and Nitrogen Species and Glutathione Redox Potential in Skeletal Muscle Cells

**DOI:** 10.3390/ijms221910876

**Published:** 2021-10-08

**Authors:** Escarlata Fernández-Puente, Jesús Palomero

**Affiliations:** 1Department of Physiology and Pharmacology, Faculty of Medicine, Campus Miguel de Unamuno, University of Salamanca, Av. Alfonso X El Sabio, 37007 Salamanca, Spain; escarlatafdezpuente@usal.es; 2Institute of Neurosciences of Castilla y León (INCyL), 37007 Salamanca, Spain; 3Institute of Biomedical Research of Salamanca (IBSAL), 37007 Salamanca, Spain

**Keywords:** biosensors, quantitative fluorescence microscopy, redox signaling, RONS, hydrogen peroxide, nitric oxide, glutathione redox potential, skeletal muscle, single skeletal muscle fiber, C2C12 myoblast/myotube

## Abstract

Reactive oxygen and nitrogen species (RONS) play an important role in the pathophysiology of skeletal muscle and are involved in the regulation of intracellular signaling pathways, which drive metabolism, regeneration, and adaptation in skeletal muscle. However, the molecular mechanisms underlying these processes are unknown or partially uncovered. We implemented a combination of methodological approaches that are funded for the use of genetically encoded biosensors associated with quantitative fluorescence microscopy imaging to study redox biology in skeletal muscle. Therefore, it was possible to detect and monitor RONS and glutathione redox potential with high specificity and spatio-temporal resolution in two models, isolated skeletal muscle fibers and C2C12 myoblasts/myotubes. Biosensors HyPer3 and roGFP2-Orp1 were examined for the detection of cytosolic hydrogen peroxide; HyPer-mito and HyPer-nuc for the detection of mitochondrial and nuclear hydrogen peroxide; Mito-Grx1-roGFP2 and cyto-Grx1-roGFP2 were used for registration of the glutathione redox potential in mitochondria and cytosol. G-geNOp was proven to detect cytosolic nitric oxide. The fluorescence emitted by the biosensors is affected by pH, and this might have masked the results; therefore, environmental CO_2_ must be controlled to avoid pH fluctuations. In conclusion, genetically encoded biosensors and quantitative fluorescence microscopy provide a robust methodology to investigate the pathophysiological processes associated with the redox biology of skeletal muscle.

## 1. Introduction

Skeletal muscle is continually exposed to several disturbances (metabolic, mechanical, functional, pathological, etc.) that affect the morphology and function of this system. However, skeletal muscle is capable of adapting to those potential injured situations with plasticity, since this organ experiences metabolic and structural transformation that provides input to resist and overcome subsequent insults and to maintain integrity and homeostasis [1,2]. The adaptation and other functions of skeletal muscle (regeneration, metabolism, and mechanical activity) are driven by different signaling pathways taking place intracellularly, and different actors are involved, such as receptors, lipids, enzymes, and second messengers, among others. 

During the last 20 years, studies in the field of redox biology have provided evidence that reactive oxygen and nitrogen species (RONS) are involved in the regulation and modulation of intracellular signaling pathways, which drive metabolism, regeneration, and sarcopenia (loss of skeletal muscle mass and force during aging) [3,4,5]. The main reactive oxygen species (ROS) in cells are free radicals such as superoxide anion and hydroxyl radical and other species that are not considered free radicals such as hydrogen peroxide (H_2_O_2_). The predominant reactive nitrogen species (RNS) in cells are nitric oxide (NO), which is a free radical generated from the amino acid L-arginine through nitric oxide synthases (NOS) enzymes [4], as well as peroxynitrite anion (ONOO-), which is generated by the reaction of superoxide and nitric oxide at diffusion-controlled rates [6]. Peroxynitrite is a short-lived oxidant, and, although it is not a free radical, it is highly reactive with thiol groups, proteins, and DNA, damaging these molecules [4,6]. ROS and RNS originate from different subcellular localizations; quantitatively, mitochondria appear to be the major source of ROS production in skeletal muscle cells [4], although there are other important sources, such as NADPH oxidases (NOXs), which are localized in the sarcolemma, sarcoplasmic reticulum, and transverse tubes in skeletal muscle cells [4,7]. Another source of ROS generation is phospholipase A_2_; the activation of this enzyme in skeletal muscle cells stimulates the production of ROS in mitochondria and cytosol and facilitates the release of ROS into the extracellular space [4,8]. Furthermore, it has been reported that xanthine oxidase generates superoxide in skeletal muscle [4,9]. Nitric oxide is mainly synthesized by a group of isoenzymes, nitric oxide synthases (NOS), which produce nitric oxide by converting the amino acid L-arginine into L-citrulline, in which NADPH and oxygen act as cofactors [4,10]. Cells are equipped with enzymatic and nonenzymatic antioxidant defense systems. The main antioxidant enzymes are superoxide dismutase, glutathione peroxidase, and catalase. In addition, there are other accessory proteins reductases that protect cells from oxidation, such as peroxiredoxin, glutaredoxin, and thioredoxin. The non-enzymatic antioxidant defense system of cells consists of various compounds such as glutathione, vitamins C and E, carotenoids, uric acid, and bilirubin [4]. Cells need to preserve a delicate balance between the formation and removal of ROS and RNS to maintain an adequate redox state that allows to carry out its functions. Any dysfunction of the antioxidant defense system may cause disturbance of the redox state. Thus, an increase in ROS production or a decrease in the capacity of the antioxidant system to neutralize ROS production lead to an increase in the intracellular level of ROS. This may drive disruption of cellular redox homeostasis and result in a phenomenon known as oxidative stress. Excess ROS may attack cellular structures such as lipids, proteins, and DNA, leading to irreversible changes in them, which can lead to block cell functions and can disrupt cellular integrity. Under normal physiological conditions, the reactive nature of RONS allows the incorporation of these reversible changes into the structure of cellular macromolecules. These reversible oxidative modifications play a crucial role in cellular signaling pathways that modulate cellular functions and cell fate [11,12].

Nowadays, it is known that RONS play an important role in the pathophysiology of skeletal muscle. However, the molecular mechanisms, the amount and specific RONS required for activation, or the contrary inhibition of pathways remain unknown. Thus, there is a need for research in the field of redox signaling to understand the molecular mechanisms responsible for the pathophysiology of skeletal muscle, and moreover, to develop pharmacological, functional, and even nutritional interventions to overcome pathologies of skeletal muscle, including those associated with aging, such as sarcopenia or fragility.

The study of RONS is particularly difficult due to the inherent properties of these molecules; they have high reactivity, non-specific reactions, and a short half-life [13]. Traditional techniques are ineffective in detecting the formation of RONS in real time and identifying the location where they are generated, since these techniques involve the processing of biological samples, and this can generate artifacts caused by sample processing. On the contrary, traditional techniques are ineffective in detecting low concentrations of RONS, which are those that modulate cell signaling pathways. On other occasions, the analysis of RONS is based on the detection of end products of reactions. There are other techniques that measure the formation of RONS in subcellular organelles isolated from their microenvironment, which inevitably causes modifications of their physiological environment. Therefore, to investigate and discern the role that RONS play in different cellular functions, it is necessary to use new more efficient methodologies. One of these methodologies is quantitative fluorescence microscopy associated with image analysis of specific detectors (biosensors) for different RONS that are targeted in different cell compartments. This methodology allows the detection of the formation of specific RONS in situ and in real time, and avoids minimal perturbation, maintaining cellular integrity and the extracellular environment. In addition, this methodology permits to quantify the production of RONS and to determine the redox state of cells. This spatio-temporal resolution, together with the monitoring of RONS generation and cellular redox status, provides an excellent methodological approach to investigate the redox biology of cells [14,15]. However, there are certain technical–methodological problems that limit the use of this methodology. One of them is that there are few detectors with high specificity and sensitivity to detect RONS that can specifically discriminate the different RONS that are generated at low concentrations. Another problem is the internalization of the detectors in cells, which may affect the functional integrity and redox state of these cells. Moreover, there are few physiological models that allow the results to be extrapolated to the real physiological context of the organism. 

With the issue of investigating the redox biology of skeletal muscle in pathophysiological situations, we undertook methodological approaches based on the use of genetically encoded biosensors to detect and monitor hydrogen peroxide, nitric oxide, and glutathione redox potential in the models of isolated mature skeletal muscle fiber and C2C12 myoblast/myotube, combined with fluorescence microscopy analysis. The integration, combination, and optimization of these different methodologies in our experimental models permit the detection and quantification, in situ and in real time, of intracellular hydrogen peroxide, nitric oxide, and glutathione redox potential in skeletal muscle cells.

## 2. Results

Different biosensors for the detection of hydrogen peroxide, nitric oxide, and glutathione redox potential were expressed in muscle cells, mainly skeletal muscle fibers and myoblasts, and the validation of the functionality of these detectors was assessed in live cells under different experimental conditions.

### 2.1. Hydrogen Peroxide Biosensor HyPer3 for the Detection of Cytosolic Hydrogen Peroxide in Skeletal Muscle Fibers

HyPer3 was expressed in the cytosol of skeletal muscle fibers maintained at 37 °C in a 5% CO_2_ atmosphere. The 5% CO_2_ atmosphere was disrupted for 40 s at two timepoints, 15–16 and 45–46 min, to approximately 0% CO_2_. This fluctuation of CO_2_ affected to fluorescence of the HyPer3 biosensor. Thus, the relative fluorescence (F/F0), emission 520/excitation 488, increased in the following 10 min after CO_2_ disruption and recovery to basal level (Figure 1a, top panel). Similarly, an increase in the lower magnitude was observed in the case of relative fluorescence (F/F0), emission 520/excitation 420 (Figure 1a, middle panel). The fluorescence rate, which is the rate of fluorescence emission 520 (excitation 488) divided by fluorescence emission 520 (excitation 420), is another way to present the fluorescence of biosensor HyPer3. In this experiment, the fluorescence rate increased as a consequence of CO_2_ fluctuations (Figure 1a, bottom panel). 

Fibers expressing HyPer3 were exposed to H_2_O_2_ and DTT. These reagents, which had been previously prepared in concentrated solutions, were added to the fiber medium to reach the final concentration indicated in every experiment, and this operation involved disruption of the CO_2_ atmosphere from 5% to 0%, since the lid that covered the CO_2_ chamber must be opened to obtain access for adding the reagents into the cell medium. When H_2_O_2_ was incorporated in the medium, the relative fluorescence (emission (emis.) 520, excitation (exc.) 488) increased in a biphasic manner, and when DTT was added, the relative fluorescence decreased dramatically 4 min after DTT addition (Figure 1b, top panel). In the case of relative fluorescence (emis. 520, exc. 420), the addition of H_2_O_2_ evoked a decrease in fluorescence 18 min after a spike of H_2_O_2_, and DTT produced an increase in fluorescence 5 min after the addition (Figure 1b, middle panel). The presentation of the fluorescence rate showed that H_2_O_2_ produced a biphasic increase and DTT evoked a dramatic decrease in the fluorescence rate (Figure 1b, bottom panel).

### 2.2. Mitochondrial Hydrogen Peroxide Biosensor HyPer-Mito for the Detection of Mitochondrial Hydrogen Peroxide in Skeletal Muscle Fibers

The hydrogen peroxide biosensor HyPer-mito was expressed in the mitochondria of skeletal muscle fibers that were maintained at 37 °C in a 0% CO_2_ atmosphere. The addition of hydrogen peroxide (100 µM) to the fiber culture medium at the 15 min timepoint evoked a dramatic increase in HyPer-mito fluorescence 5 min later. This indicates that hydrogen peroxide had diffused from the extracellular space to the cytosol, and once in the cytosol, H_2_O_2_ diffused into mitochondria matrix, where it reacted with the HyPer-mito biosensor, which produced the increase in fluorescence. The increase in fluorescence in the mitochondria was transient, reaching the peak and maintaining it for 5 min, after which the fluorescence decreased to the basal level. At this time (39 min), the fiber showed constriction—a signal of toxicity (Figure 2a). 

We undertook an experiment to prove whether CO_2_ might affect the fluorescence of the HyPer-mito biosensor. A fiber expressing the mitochondrial biosensor HyPer-mito was maintained at 37 °C in a 0% CO_2_ atmosphere, and fluorescence was monitored. At the 40 min timepoint, the CO_2_ atmosphere was exchanged from 0% to 5% CO_2_ and fluorescence was monitored for other 80 min. The fluctuation from 0% to 5% CO_2_ atmosphere produced a decrease in HyPer-mito fluorescence, and the fiber appeared viable without any signal of toxicity at the end of the time course (120 min) (Figure 2b).

### 2.3. Nuclear Hydrogen Peroxide Biosensor HyPer-nuc for the Detection of Nuclear Hydrogen Peroxide in Skeletal Muscle Fibers

We expressed hydrogen peroxide biosensor HyPer-nuc in the nuclei of skeletal muscle fibers. The fibers were maintained at 37 °C in a 0% CO_2_ atmosphere for the initial 15 min, followed by 75 min in a 5% CO_2_ atmosphere. When the CO_2_ atmosphere switched from 0% to 5% CO_2_, the fluorescence of HyPer-nuc decreased, either green fluorescence (emis. 520, exc. 488) or blue fluorescence (emis. 520, exc. 420), although the fluorescence rate remained constant and decayed after 6 min. The addition of H_2_O_2_ (50 µM) to the fiber medium, in a 5% CO_2_ atmosphere, did not produce any change in nuclear fluorescence initially. However, after 15 min, green fluorescence (emis. 520, exc. 488) increased, blue fluorescence (emis. 520, exc. 420) decreased, and this resulted in the rate of fluorescence dramatically increasing. The addition of reductant agent DTT at 76 min immediately produced (in 1 min) a decrease in green fluorescence, an increase in blue fluorescence, and a dramatic decline in the fluorescence rate. However, the fibers hypercontracted 2 min after the addition of DTT, probably due to the toxicity evoked by this high reductant capacity of DTT (Figure 3).

### 2.4. Hydrogen Peroxide Biosensor roGFP2-Orp1 for the Detection of Cytosolic H_2_O_2_ in Skeletal Muscle Fibers

Hydrogen peroxide biosensor roGFP2-Orp1 was expressed in the cytosol of skeletal muscle fibers and maintained at 37 °C in a 5% CO_2_ atmosphere for a 60 min time course. At the 10 min timepoint, H_2_O_2_ was added to the fiber medium. This operation involved disruption of the CO_2_ atmosphere from 5% to 0% CO_2_ for 40 s. The addition of H_2_O_2_ produced a rapid increase in green fluorescence (emis. 520, exc. 488) for 2 min, followed by a decrease in fluorescence that remained constant and under the basal fluorescence level for 35 min. However, blue fluorescence (emis. 520, exc. 420) appeared to remain constant without appreciable changes. The expression of results as the fluorescence rate, blue fluorescence (emis. 520, exc. 420)/green fluorescence (emis. 520, exc. 488), showed an increase in the fluorescence rate as consequence of the addition of H_2_O_2_. When the reductant reagent DTT was added to the fiber medium, either the green or blue fluorescence increased dramatically, and the fluorescence rate showed decay. Ten minutes after the addition of DTT, the fibers were hypercontracted, probably due to DTT exposure, since DTT might reduce the disulfide bonds of structural proteins and denaturant them, which might affect the structure and morphology of the fiber (Figure 4a). Similarly, when insulin was added to the fiber medium, this produced a rapid increase in either green fluorescence or blue fluorescence; however, the fluorescence rate did not change (Figure 4b). These fluorescence changes appear to be due to the effect of pH, since adding insulin involved the disruption of the CO_2_ atmosphere from 5% to 0% CO_2_ for 40–50 s, and this might have evoked a rise in pH, which affected the fluorochrome domine (roGFP2) of the roGFP2-Orp1 biosensor and then produced the increase in fluorescence.

### 2.5. Mito-Grx1-roGFP2 Biosensor to Register Mitochondrial Glutathione Redox Potential (GSH/GSSG) in Myoblasts

The mito-Grx1-roGFP2 biosensor was expressed in C2C12 myoblasts for monitoring the glutathione redox potential (GSH/GSSG) in the mitochondrial matrix of these cells. Figure 5 presents fluorescence microscopy images that show that mito-Grx1-roGFP2 is expressed specifically in the mitochondria of C2C12 myoblasts (images I–IV). The addition of H_2_O_2_ to the cell medium (37 °C and 0% CO_2_) evoked a decrease in the biosensor green fluorescence (emis. 520, exc. 488) 5 min after the addition of H_2_O_2_ and then remained constant and below basal fluorescence level. In parallel, blue fluorescence (emis. 520, exc. 420) remained constant. This indicates that the fluorescence rate, blue fluorescence (emis. 520, exc. 420)/green fluorescence (emis. 520, exc. 488), showed an increase due to the addition of H_2_O_2_. The addition of DTT to the cellular medium evoked an increase in green fluorescence, no changes in blue fluorescence, and a consequent decrease in the fluorescence rate. The exposure of cells to oxidants (H_2_O_2_) produced an increase in the fluorescence rate, meaning an increase in GSSG and then a decrease in GSH/GSSG (glutathione redox potential). By contrast, DTT, a strong reductant, produced a reduction of GSSG to GSH and then an increase in GSH/GSSG.

### 2.6. Cyto-Grx1-roGFP2 Biosensor to Register Cytosolic Glutathione Redox Potential (GSH/GSSG) in Myotubes

The cyto-Grx1-roGFP2 biosensor was expressed in C2C12 myoblasts, and then these cells were experimentally induced to differentiate into myotubes. Cyto-Grx1-roGFP2 is a cytosolic biosensor for monitoring the glutathione redox potential (GSH/GSSG) in the cytosol. However, although the myotubes expressed the cyto-Grx1-roGFP2 biosensor (Figure 6), the biosensor did not react under experimental conditions, positive controls, which might have potentially evoked changes in glutathione redox potential, i.e., exposure to H_2_O_2_.

### 2.7. G-geNOp Biosensor to Detect Cytosolic Nitric Oxide in Isolated Skeletal Muscle Fibers

The nitric oxide biosensor G-geNOp was expressed in the cytosol of skeletal muscle fibers that were maintained at 37 °C in a 0% CO_2_ atmosphere for a 60 min time course. The nitric oxide donor S-nitroso-N-acetylpenicillamine (SNAP) was added into the fiber medium at different timepoints (10, 20, and 31 min), and the green fluorescence (emis. 520, exc. 488) of the G-geNOp biosensor, expressed in the fiber cytosol, decayed dramatically. This effect was due to the NO released from SNAP into the medium, when the NO diffused through sarcolemma, and once in the cytosol this NO reacted with G-geNOp, which produced quenching of the biosensor green fluorescence (emis. 520, exc. 488) (Figure 7a). Figure 7b presents the monitoring of green fluorescence (emis. 520, exc. 488) from a skeletal muscle fiber that expressed the biosensor G-geNOp for a 60 min time course. The first 20 min fiber was incubated at 37 °C in a 5% CO_2_ atmosphere. Under these conditions, green fluorescence from the fibers maintained constant. However, when the CO_2_ atmosphere was exchanged from 5% to 0% CO_2_, an increase in green fluorescence commenced softly and progressively.

## 3. Discussion

The evidence regarding the role of RONS in signaling processes is supported by many studies using different models [3]. However, it is still difficult to know which RONS drive each process, in addition to the quantity of RONS, and furthermore, when and where RONS are generated. In other words, the dynamics of RONS in cells and intracellular organelles and the interplay between cells and extracellular fluids. Different methodologies and technical approaches have been used to determine, at least indirectly, the involvement of RONS in different signaling pathways [3]. Fifteen years ago, a new cornerstone arose that may have revolutionized the field of redox biology in terms of the analysis, quantitation, and location of intracellular RONS. This is the advent of genetically encoded biosensors, which are capable of detecting specific RONS and redox states in subcellular domines. One of these biosensors is HyPer, a hydrogen peroxide biosensor, developed in and available since 2006 [16]. This biosensor has been improved, and nowadays, there is a family of HyPer biosensors with characteristics suitable for different models [17]. Some of them are skeletal muscle models [18,19,20,21]. In this study, we reported the results in which biosensor HyPer3 was expressed in mature skeletal muscle fibers. HyPer3 presented an improved performance for detecting hydrogen peroxide compared to its predecessors HyPer and HyPer2. Thus, HyPer3 responds with a higher dynamic range and a quick response when it reacts with hydrogen peroxide [22]. In our study, HyPer3 was expressed in mouse mature skeletal muscle fibers, and it detected the intracellular changes of H_2_O_2_ and the high reductant state induced by DTT (Figure 1b). In addition, we confirmed that the biosensor HyPer3 was sensitive to pH, since changes in environmental CO_2_ may have affected the pH of the medium of the fibers, and this was reflected in the fluorescence of the biosensor. Thus, in the absence of CO_2_, the pH of the medium increased (basic pH) and the fluorescence of HyPer3 increased as well. However, when the CO_2_ atmosphere returned to 5% CO_2_, the pH of the medium decreased (acidic) and the fluorescence of HyPer3 decreased as well (Figure 1a). It has been reported that HyPer is sensitive to pH, since the fluorescence domain of the biosensor, cpYFP, is sensitive to pH [23]. The sensitivity of HyPer to pH has been considered a disadvantage of this kind of biosensor, since the fluorescence changes produced by the effect of pH might mask the fluorescence changes produced by H_2_O_2_ or reductant ambient [14]. However, we think that this could be an additional characteristic of this biosensor and may be overcome by considering and controlling the CO_2_ environment of cells around 5%, which may regulate the pH of the fiber medium around neutral pH.

The hydrogen peroxide biosensor HyPer may be targeted to subcellular compartments, such as mitochondria and nuclei [24]. We expressed the biosensor HyPer-mito in the mitochondria of skeletal muscle fibers, then carried out time course experiments to monitor the fluorescence of this biosensor, which allowed to detect the presence of hydrogen peroxide into the mitochondria after isolated fibers in the culture had been exposed to extracellular H_2_O_2_, which had been added to the fiber culture medium (Figure 2a). This experiment may be considered a positive control for the detection of H_2_O_2_ in the mitochondria. The fluorescence of HyPer-mito revealed that H_2_O_2_, which was added extracellular to the fiber medium, diffused from the extracellular space to the cytosol and then to the mitochondrial matrix. Once H_2_O_2_ reached the mitochondrial matrix, it reacted with the HyPer-mito biosensor and this produced an increase in fluorescence. Simultaneously and in competition, H_2_O_2_—which was internalized into the mitochondria—was neutralized by the antioxidant system in the mitochondrial matrix, since the fluorescence of HyPer-mito decreased, and finally, the fiber hypercontracted due to toxicity (Figure 2a). These results support that the biosensor HyPer-mito may be an adequate biosensor for monitoring the presence of H_2_O_2_ in the mitochondria in real time in skeletal muscle fibers. Even more, this positive control experiment established the fundamentals for undertaking further studies in the future, in which the production of H_2_O_2_ in the mitochondria will be stimulated (using mitochondria electronic transport blockers, such as antimycin A and rotenone) and the HyPer-mito biosensor will be used to monitor and assess the generation of H_2_O_2_ in the mitochondrial matrix. Although HyPer-mito has been used in different models such as pancreatic beta-cells [25] and embryonic fibroblast cell line NIH 3T3 [24] to monitor mitochondrial H_2_O_2_, there is only one study carried out in skeletal muscle fibers, where HyPer-mito was used to monitor H_2_O_2_ and to study the interplay between H_2_O_2_/superoxide/mitochondrial pH [25]. The HyPer-mito biosensor is sensitive to pH changes originated by ambient CO_2_ fluctuations in the fiber medium (Figure 2b, bottom panel). This is due to the fluorescence of HyPer, which might be pH-dependent, since the fluorescence emitted by the fluorochrome domain of HyPer, cpYFP, is pH-dependent [14]. This suggests that it is critical to control the CO_2_ environment of live cells to avoid pH fluctuations that might evoke fluorescence changes in HyPer-mito, which might confound the registration of biological changes and mask the real results.

There is a version of HyPer, HyPer-nuc, that may target the nucleus [26]. We expressed HyPer-nuc in the nuclei of skeletal muscle fibers (Figure 3) and demonstrated that this biosensor detects changes in hydrogen peroxide in the nucleus after external addition of hydrogen peroxide to the fiber medium, indicating that hydrogen peroxide crossed sarcolemma, diffused into the nucleus, and then reacted with the HyPer biosensor, producing an augmentation of fluorescence. Similar to other versions of HyPer, HyPer-nuc is reduced by DTT, and it is sensitive to pH changes in the cellular medium evoked by CO_2_ ambient fluctuations. Therefore, HyPer-nuc may be used in skeletal muscle fibers to detect and, in some way, quantify H_2_O_2_ in the nuclei of fibers. This may provide information for knowing and understanding the hydrogen peroxide flux in skeletal muscle fibers. 

roGFP2-Orp1 is another genetically encoded biosensor for hydrogen peroxide detection that was developed by Gutscher and collaborators [27]. This biosensor is an engineered fusion protein constituted by a redox-sensitive green fluorescence protein (roGFP2) linked to a yeast peroxidase (Orp1) [27]. Orp1 mediates the oxidation of roGFP2 by H_2_O_2_ and produces changes in the fluorescence emission of fluorochrome GFP2. Therefore, the expression of the roGFP2-Orp1 biosensor in living cells enables the measurement of intracellular H_2_O_2_ in a specific and sensitive manner under physiological conditions [27,28,29]. The roGFP2-Orp1 biosensor has been used in in vitro models and expressed in cell line models, such as HeLa cells and primary human peripheral blood mononuclear cells, where the coding sequence of roGFP2-Orp1 was internalized into cells by transfection techniques [27], and in other study using transduction techniques, where the coding sequence of roGFP2-Orp1was packed in adenovirus to express the biosensor in pancreatic beta cells from rat [30]. However, as far as we know, roGFP2-Orp1 has not been expressed in skeletal muscle fibers, which is the model undertaken in our study (Figure 4). We expressed roGFP2-Orp1 in mature skeletal muscle fibers using a microinjection of the coding sequence of roGFP2-Orp1 in the FDB muscle, followed by electroporation in this muscle. We demonstrated that the roGFP2-Orp1 biosensor expressed in skeletal muscle fibers detects intracellular H_2_O_2_ flux and is affected by the reduction produced by DTT (Figure 4a). In addition, it was shown that roGFP2-Orp1 is sensitive to pH, and the fluorescence was affected when CO_2_ atmosphere was disrupted from 5% to 0% CO_2_. This is the case after the addition of reagents, i.e., H_2_O_2_ and DTT (Figure 4a) or insulin (Figure 4b). It appears that insulin does not stimulate the production of H_2_O_2_. This is in contrast to the increase in H_2_O_2_ produced by insulin in skeletal muscle fibers from mice on a high-fat diet that expressed hydrogen peroxide biosensor HyPer [18].

The concern regarding the sensitivity to pH changes of hydrogen peroxide biosensors HyPer and roGFP2-Orp1 has been pointed out in different publications [27,28,29,31,32]. Our results are in accordance with this and revealed that the hydrogen peroxide biosensors HyPer3 and roGFP2-Orp1 are pH-sensitive. Since the pH of the cell culture medium is affected by environmental CO_2_ (Appendix A) and the fluorescence emitted by biosensors may depend on pH fluctuations, this must be considered when experiments are planned and it will be necessary and critical to implement models that maintain pH by regulation of ambient CO_2_ and temperature, which might avoid pH fluctuations in the cell culture medium. Otherwise, the fluorescence emitted by biosensors might be artefactual and mask the real biological signal—in this case, the flux of H_2_O_2_ in skeletal muscle fibers.

Glutathione (GSH) homeostasis constitutes the non-enzymatic antioxidant cellular system that is responsible for maintaining the integrity of cells, since GSH is the main regulator of the intracellular redox state and it neutralizes ROS and RNS produced in excess [4,33,34]. There are different techniques, mainly spectrophotometric and fluorometric, to quantify reduced glutathione (GSH) and oxidized glutathione (GSSG) and then estimate the redox potential of glutathione (GSH/GSSG), which, in some way, indicates the proportion of GSSG in the pool of total GSH in tissues such as skeletal muscle [35]. The traditional methodology to determine GSH involves the processing of samples and end-point reactions that determine the GSH and GSSG content in the samples. Therefore, the results obtained with this methodology reflect the intracellular redox state at one temporal point. However, there is not information regarding the course of a process where cells or tissues are subjected; in other words, there is not temporal resolution of the cellular redox state. In 2008, Gutscher and collaborators created a biosensor to detect intracellular GSH/GSSG called Grx1-roGFP2 [36]. The biosensor Grx1-roGFP2 is a fusion protein that provides dynamic images of the glutathione redox potential in cells with temporal resolution and high sensitivity [36]. There is not sample processing, nor disruption of cellular integrity, and the determination of GSH/GSSG is caried out in live cells at different timepoints during a time course. Therefore, this biosensor indicates GSH/GSSG during a time course where cells are subjected to different agents that may affect the redox state. Moreover, the biosensor Grx1-roGFP2 may be targeted toward different subcellular compartments, such as cytosol [36] and mitochondria [37]. Several approaches have been undertaken to express the biosensor Grx1-roGFP2 in cell lines [36] and tissues [38], and even transgenic organisms expressing Grx1-roGFP2 have been created [39]. As far as we know, to date, there is not any publication in the literature that refers to any model of skeletal muscle where the biosensor Grx1-roGFP2 would have been used to determine GSH/GSSG. We achieved the expression of the biosensor Grx1-roGFP2 in the cytosol of myotubes C2C12, a skeletal muscle cellular model (Figure 6). However, the biosensor cyto-Grx1-roGFP2 did not respond to the exposure of reagents, such as H_2_O_2_, that potentially may affect glutathione redox potential. This inconsistency might be explained considering that the sequence of the biosensor might be affected by any mutation that might alter the functionality of the biosensor but not the fluorescence. On the contrary, we expressed mito-Grx1-roGFP2, a mitochondrial targeted version of biosensor Grx1-roGFP2, in the mitochondria of myoblasts C2C12, a well-known skeletal muscle cell line (Figure 5). In this case, we proved that the mitochondrial biosensor mito-Grx1-roGFP2 responded and registered changes in the mitochondrial redox potential of glutathione produced by an oxidant reagent, H_2_O_2_, and a reductant reagent, DTT. Demonstrating that this biosensor is capable of monitoring the mitochondrial redox potential of glutathione in real time in the mitochondria from live myoblasts.

Nitric oxide (NO) is a reactive nitrogen species (RNS) with a prominent role in different organism functions. Nitric oxide is constantly produced in skeletal muscle and it is involved in several processes in this organ such as blood flow, contractile activity, passive stretching, glucose homeostasis, regeneration, and aging [2,40,41,42,43]. Different methodologies have been used to detect and quantify extracellular and intracellular nitric oxide [44]. However, there are limitations to obtaining reliable results, since biological samples need to be processed, and this might generate artefacts that may influence the acquisition of consistent results. Moreover, the results obtained are based on end-point reactions and restricted to one timepoint. Therefore, there is a lack of spatial and temporal information regarding the flux of nitric oxide in cells. In 2006, Eroglu and collaborators developed new probes for the detection and monitoring of NO in living cells. These NO detectors are chimeric proteins genetically encoded in constructs formed by a NO binding domain linked to a fluorescence protein [15]. We used one of these NO detectors, G-geNOp, in our study (Figure 7). To the best of our knowledge, this is the first time that a genetically encoded NO biosensor has been used in skeletal muscle cells. G-geNOp is a chimeric construct, formed by a variant of GFP that is fused with a highly selective non-heme iron(II) NO binding domain. NO binds to G-geNOp, inducing a loss in fluorescence intensity, which is completely recovered when NO is dissociated from G-geNOp [45].

We expressed the G-geNOp biosensor in mouse skeletal muscle fibers to evaluate the intracellular NO flux in real time using fluorescence microscopy imaging. Previously, the plasmid with the codding sequence of G-geNOp was microinjected and electroporated into FDB muscle, followed by fiber isolation after four days. We assayed the addition of a NO donor, SNAP, to the medium of fibers that expressed G-geNOp and monitored intracellular NO flux during a time course. The fluorescence signal of G-geNOp demonstrated that after the addition of SNAP, NO crossed the sarcolemma and was internalized into the fibers, where, after the reaction with the biosensor, the fluorescence was quenched, indicating the presence of NO in the cytosol of the fibers (Figure 7a). Furthermore, we proved that the fluorescence of the NO biosensor G-geNOp was affected by pH. Thus, when the CO_2_ atmosphere concentration was 5%, this resulted in a neutral pH in the fiber medium, and then the fluorescence of G-geNOp remained constant. However, when the CO_2_ atmosphere concentration exchanged to 0%, that means that the pH of fiber medium changed to basic, and then the fluorescence of the biosensor G-geNO increased (Figure 7b). In summary, we achieved the expression of the NO biosensor G-geNOp in isolated skeletal muscle fibers. The application of fluorescence microscopy imaging methodology in isolated skeletal muscle fibers, that express the NO biosensor G-geNOp, is appropriate for detecting and monitoring in real time the flux of NO in skeletal muscle. The G-geNOp biosensor is susceptible to pH fluctuations, which may influence the biosensor fluorescence emission and might mask the fluorescence produced by the reaction of NO with G-geNOp. Thus, the experimental conditions of fibers, such as environmental CO_2_, must be considered to interpretate results in a reliable manner. When the biosensor G-geNOp is used to detect intracellular NO, the CO_2_ environment of the cells must be controlled to maintain pH and avoid misinterpretation of the results, which might be underestimated or overestimated. This suggests that there is a need to introduce methodological improvements that avoid CO_2_ fluctuations in order to assess accurate intracellular NO in live cells. Finally, the property of G-geNOp as a detector of pH changes should be explored, since this biosensor might be useful for situations with environmental CO_2_ fluctuations, which might be evoked by gases, such as oxygen (O_2_) and nitrogen (N_2_).

## 4. Materials and Methods

### 4.1. Reagents 

The hydrogen peroxide solution (30 wt% in water), dithiothreitol (DTT), S-nitroso-N-acetylpenicillamine (SNAP), and recombinant human insulin were obtained from Merck-Sigma-Aldrich (Darmstadt, Germany).

### 4.2. Skeletal Muscle Cell Culture 

The C2C12 mouse skeletal myoblasts were purchased from the American Type Culture Collection (CRL-1772, Manassas, Virginia, USA). C2C12 was cultured in Dulbecco’s modified Eagle medium (DMEM; Sigma-Aldrich) containing 10% (*v*/*v*) fetal bovine serum (FBS), (Invitrogen, Waltham, Massachusetts, USA), 2 mM L-glutamine (Sigma-Aldrich), 50 i.u. penicillin, and 50 μg of mL−1 streptomycin (Sigma-Aldrich) at 37 °C in an atmosphere of 5% CO_2_ and adequate humidity. C2C12 myoblasts were differentiated for 7 days to obtain myotubules; during this time, the myoblasts fused and transformed into multinuclear myotubes with a tubular morphology. Differentiation involved growing myoblasts to 80%–90% confluency and then replacing the growth medium with differentiation medium (DMEM with 2% (*v*/*v*) horse serum (Invitrogen) supplemented with 2 mM L-glutamine, 50 i.u. penicillin, and 50 μg mL^−1^ of streptomycin) [21]. 

### 4.3. Animals

Female and male 3–5 month-old C57BL/6J mice were used in this study. The procedures involving animals were approved by the Bioethics Committee of the University of Salamanca and conducted in accordance with the Spanish (RD 53/2013) and European Union (2010/63/EU) guidelines for animal experimentation.

### 4.4. Genetically Encoded Biosensors

Vectors (plasmids) with the coding sequence of biosensors were obtained from repository Addgene (Watertown, Massachusetts, USA) and from commercial sources. HyPer3 is a fluorescence biosensor used to detect intracellular hydrogen peroxide [22]; the DNA sequence (pC1-HyPer-3) was acquired from Addgene, a gift from Vsevolod Belousov (Addgene plasmid # 42131). The DNA sequences of HyPer-mito (pHyPer-dMito vector) and HyPer-nuc (pHyPer-nuc vector), which are mammalian expression vectors encoding mitochondria or nuclear-targeted HyPer, respectively, were purchased from the company Evrogen (Moscow, Russia). The DNA sequence (pLPCX roGFP2-Orp1) of roGFP2-Orp1, a biosensor that measures intracellular hydrogen peroxide [27], was acquired from Addgene, a gift from Tobias Dick (Addgene plasmid # 64991). The DNA sequences of cyto-Grx1-roGFP2 (pLPCX cyto Grx1-roGFP2) and mito-Grx1-roGFP2 (pLPCX mito Grx1-roGFP2), biosensors that detect changes in oxidized glutathione (GSSG) and glutathione redox potential (GSH/GSSG) in the cytosol and mitochondria [36], were acquired from Addgene, a gift from Tobias Dick (Addgene plasmids # 64975 and # 64977). G-geNOp is a green fluorescence-based probe for the detection of intracellular nitric oxide [15], and the DNA sequence of G-geNOp was packed in a plasmid that was purchased from the company Next Generation Fluorescence Imaging (NGFI, Graz, Austria).

DNA vectors with the sequence of biosensors were amplified in DH5alfa *E. coli*, followed by plasmid purification using a Plasmid Midiprep Kit (GeneJET, Thermo Scientific, Waltham, Massachusetts, USA) to obtain the amount of DNA required for transfection and microinjection. The biosensor vectors HyPer-3, HyPer-mito, HyPer-nuc, and roGFP2-Orp1 were microinjected into the FDB muscle, followed by electroporation in that muscle, which produced the expression of the biosensors in muscle fibers. In the case of C2C12 myoblasts, the biosensor vectors cyto-Grx1-roGFP2, mito-Grx1-roGFP2, and G-geNOp were transfected in these cells using a method based on a polymeric transfection reagent (Viromer RED, Lypocalyx, Halle, Germany), according to the protocol optimized in our laboratory [21]. 

### 4.5. Transfection of Biosensors into Flexor Digitorum Brevis (FDB) Mouse Muscle

Microinjection of the DNA vectors HyPer-3, HyPer-mito, HyPer-nuc, and roGFP2-Orp1 was performed in FDB mouse muscle. The contralateral FDB was considered a control (no microinjection), following the protocol optimized in our laboratory [21]. Mice were maintained under gas anesthesia (3% isoflurane) during the microinjection procedure. Microinjection was carried out using a micro-syringe provided with a 0.3 mm (30G) × 8 mm needle (BD Micro-Fine+, Franklin Lakes, New Jersey, USA). First, 10 μL of sodium chloride hyaluronidase solution (2 mg/mL) (SIGMA, Darmstadt, Germany) was injected subcutaneously into the mouse pad between the FDB muscle and the skin. After 60 min of stabilization, 10 μL of the DNA coding sequence of the biosensors (biosensor plasmid) was injected into the same FDB muscle. After 15 min of stabilization, two stainless steel needle electrodes (0.3 mm (30G) × 8 mm), with a 0.5 cm separation between electrodes were placed subcutaneously and perpendicular to the FDB. Then, electroporation was applied using an electroporator system BTX-ECM830 (BTX The Electroporation Experts, Holliston, Massachusetts, USA), applying 20 square positive pulses, with a 1 s interval between pulses, a 100 V pulse potential, and a 20 ms pulse amplitude. After electroporation, the mice were allowed to recover from the anesthesia and maintained for five days until muscle fiber isolation.

### 4.6. Skeletal Muscle Fibers Isolation

Single skeletal muscle fibers were isolated from the FDB muscle from the mice according to the protocol established by Palomero and collaborators [42,46]. The isolated skeletal muscle fibers were maintained in a 5% CO_2_ humidified atmosphere at 37 °C for 24–48 h until the fluorescence microscopy experiments were undertaken.

### 4.7. Fluorescence Microscopy and Quantitative Image Analysis

C2C12 myoblasts, myotubes, and single skeletal muscle fibers that expressed the biosensors were placed under a fluorescence microscope (Live Cell Observer, Carl Zeiss, Oberkochen, Germany) equipped with a chamber to maintain temperature at 37 °C in a 0%–5% CO_2_ atmosphere to conduct experiments to assess the functionality of the biosensors in real time in these cells. The culture medium was removed from the cells and replaced by Krebs solution (119 mM NaCl, 2.5 mM KCl, 2.5 mM CaCl_2_-2H_2_O, 1.5 mM MgSO_4_-7H_2_O, 1.25 mM NaH_2_PO_4_, 26.2 mM NaHCO_3_, and 11.1 mM glucose; pH 7.4) before starting the experiments using the microscope. This Krebs solution was used for the maintenance of cells at 37 °C under a 0%–5% CO_2_ atmosphere during the sequence of the image recording time course. Images were obtained every 1 min during a time course with a 20× or a 40× magnification objective. The source of excitation light was a light-emitting diode (LED) that generated 470 nm monochromatic light. Fluorescence images were obtained through a fluorescence cube with a 450/40 nm excitation filter, 495 nm beam splitter, and 525/50 nm emission filter. Fluorescence images were acquired with a computer-controlled CCD camera (AxioCam MRm, Carl Zeiss) coupled to the microscope. The fluorescence images were acquired with the minimum exposure time that provided images with enough quality for quantification to avoid light-induced damage to the cells. Time lapse images were acquired with the same exposure time, and this was maintained for every time lapse experiment performed under the same conditions. The fluorescence image analysis was undertaken using the software (ZEN 2 blue edition, Carl Zeiss) included with the microscopy equipment. The fluorescence images recorded in every time lapse experiment were quantified. Thus, a region of interest (ROI) was selected in a cell or fiber that expressed the biosensor. In addition, another ROI was selected outside the cell or fiber to represent the background. The software provided the measurement, in terms of grey activated pixels, of the fluorescence intensity of the two ROIs. The net fluorescence of the cell or fiber expressing the biosensor was obtained by subtracting the fluorescence value of the background ROI from the fluorescence value of the cell or fiber ROI, and this represented the raw net fluorescence of the cell or fiber expressing the biosensor, which was determined at each timepoint in which an image was acquired during the time lapse experiment. On some occasions, the raw net fluorescence was transformed into the relative net fluorescence, which is more appropriate for statistical analysis, since this avoids differences in fluorescence intensity due to a potential difference in the expression of the biosensor in cells. The relative net fluorescence was calculated for each timepoint of the time lapse experiment by using the net raw fluorescence at the initiation of said experiment (rawF0) as the reference for normalization, which had a value of 1.00. The normalized fluorescence for the timepoint n (normFn) was calculated by dividing the raw fluorescence at this timepoint (rawFn) by the rawF0. Thus, normFn = rawFn/rawF0. In other occasions, fluorescence image analysis consisted of a ratiometric analysis. Thereby, fluorescence images were obtained from two channels, green and blue. The image from the green channel indicated the fluorescence emission at 520 nm under excitation with 470 nm monochromatic LED using a set of filters (450/40 nm excitation filter, 495 nm beam splitter, and 525/50 nm emission filter). The image from the blue channel indicated the fluorescence emission at 520 nm under excitation with UV light (at a range of approximately 420 nm) using a set of filters (395–440 nm excitation filter, 460 nm beam splitter, and 470 LP nm emission filter). Imaging analysis was used to determine the ratiometric fluorescence in the fibers or cells. Thus, the ratio = F emis. 520 (exc. 488)/F emis. 520 (exc. 420) for HyPer biosensors, which contain fluorochrome cpYFP, and the ratio = F emis. 520 (exc. 420) / F emis. 520 (exc. 488) for biosensors that contain fluorochrome roGFP2, where F emis. 520 (exc. 488) is the net fluorescence from the green channel and F emis. 520 (exc. 420) is the net fluorescence from the blue channel [21].

## 5. Conclusions and Future Perspectives

The use of genetically encoded biosensors for the detection of different RONS, such as hydrogen peroxide and nitric oxide, and the glutathione redox potential in the models of single skeletal muscle fiber and myoblast/myotube cell line C2C12, in combination with quantitative fluorescence microscopy imaging, is a powerful and robust methodology to study in situ and in real time redox processes in skeletal muscle. In addition, this methodology might be implemented, with specific adaptations, to other tissue models, such as cardiac muscle, particularly the isolated cardiomyocytes from heart ventricle, and in smooth muscle cells and tissue.

The fluorescence emitted by biosensors is susceptible to pH changes in the extracellular medium evoked by fluctuations in the environmental CO_2_ of cells. This may affect the correct registration of results, and consequently, these results may be masked or even underestimated or overestimated. Therefore, it is strongly recommended to control the environmental CO_2_ of cells in addition to temperature. 

Finally, the availability of genetically encoded biosensors, the improved versions, and those novel biosensors expected to come, in combination with physiological models and in situ and real time quantitative fluorescence microscopy, may uncover redox processes and help understand the redox biology and pathophysiology of organs, such as skeletal, cardiac, and smooth muscle, and potentially other systems of the organism. 

## Figures and Tables

**Figure 1 ijms-22-10876-f001:**
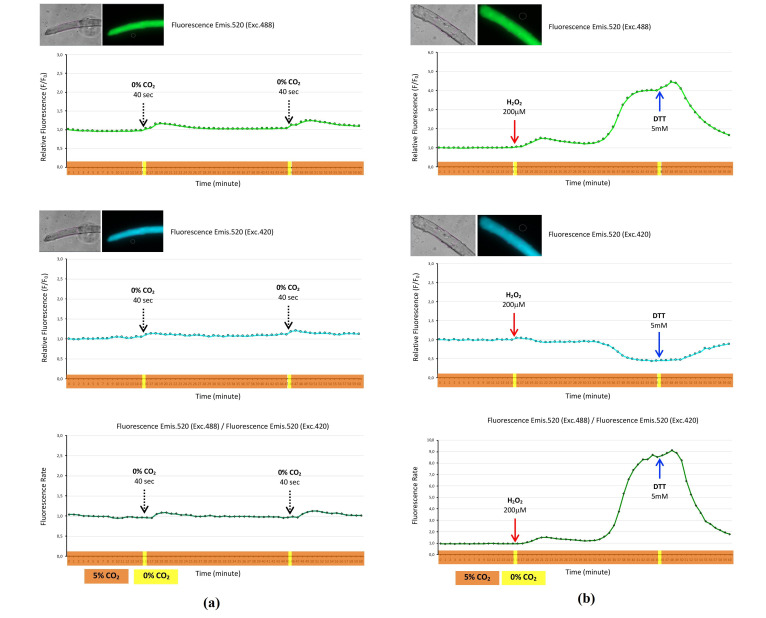
Hydrogen peroxide biosensor HyPer3 expressed in the cytosol of single skeletal muscle fibers. Fluorescence monitoring during a time course of 60 min, with fluorescence image registration every minute. Fibers were maintained at 37 °C in a 5% CO_2_ atmosphere (timeline highlighted in orange), with the exception of two timepoints at 15 and 45 min, where fibers were maintained respectively at 37 °C in a 0% CO_2_ atmosphere for 40 s (timeline highlighted in yellow). (**a**) Top panel presents the relative fluorescence (F/F0) of the HyPer3 biosensor’s fluorescence emission at 520 nm when excited at 488 nm of the fiber that expressed HyPer3. Fiber images: Transmission (grey) and fluorescence (green). Middle panel presents the relative fluorescence (F/F0) of the HyPer3 biosensor’s fluorescence emission at 520 nm when excited at 420 nm of the fiber that expressed HyPer3. Fiber images: Transmission (grey) and fluorescence (blue). Bottom panel presents the fluorescence rate of the HyPer3 biosensor, obtained from the registered fluorescence: Fluorescence emission at 520 nm when excited at 488 nm divided by the fluorescence emission at 520 nm when excited at 420 nm; (**b**) hydrogen peroxide 200 µM was added to the fiber medium at the 15 min timepoint and dithiothreitol (DTT) 5 mM at the 45 min timepoint while the fibers were at 37 °C in a 0% CO_2_ atmosphere (timeline highlighted in yellow). Top panel presents the relative fluorescence (F/F0) of the HyPer3 biosensor’s fluorescence emission at 520 nm when excited at 488 nm of the fiber that expressed HyPer3. Fiber images: Transmission (grey) and fluorescence (green). Middle panel presents the relative fluorescence (F/F0) of the HyPer3 biosensor’s fluorescence emission at 520 nm when excited at 420 nm of the fiber that expressed HyPer3. Fiber images: Transmission (grey) and fluorescence (blue). Bottom panel presents the fluorescence rate of the HyPer3 biosensor, obtained from the registered fluorescence: Fluorescence emission at 520 nm when excited at 488 nm divided by the fluorescence emission at 520 nm when excited at 420 nm. (Appendix A.)

**Figure 2 ijms-22-10876-f002:**
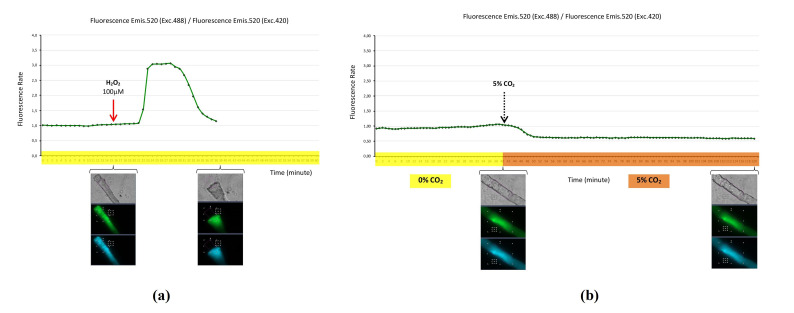
Hydrogen peroxide biosensor HyPer-mito expressed in the mitochondria of single skeletal muscle fibers. (**a**) Mitochondrial fluorescence monitoring during a time course of 60 min, with fluorescence image registration every minute. Fibers were maintained at 37 °C in a 0% CO_2_ atmosphere (timeline highlighted in yellow). Hydrogen peroxide (100 µM) was added to fiber medium at the 15 min timepoint. Fluorescence values are expressed as the fluorescence rate: Fluorescence emission at 520 nm when excited at 488 nm divided by fluorescence emission at 520 nm when excited at 420 nm. At timepoints 15 and 39 min, are two sets with three fiber images: Transmission (grey), green fluorescence (emis. 520, exc. 488), and blue fluorescence (emis. 520, exc. 420). Fibers were affected by toxicity at the 39 min timepoint and the values of fluorescence were discarded; (**b**) mitochondrial fluorescence monitoring during a time course of 120 min, with fluorescence image registration every minute. Fibers were maintained at 37 °C in a 0% CO_2_ atmosphere (timeline highlighted in yellow) during the first 40 min, following by 37 °C in a 5% CO_2_ atmosphere (timeline highlighted in orange) for 80 min. Sets of fiber images: Transmission (grey), green fluorescence (emis. 520, exc. 488), and blue fluorescence (emis. 520, exc. 420) at timepoints 40 and 120 min. Fluorescence values are expressed as the fluorescence rate: Fluorescence emission at 520 nm when excited at 488 nm divided by fluorescence emission at 520 nm when excited at 420 nm. (Appendix A.)

**Figure 3 ijms-22-10876-f003:**
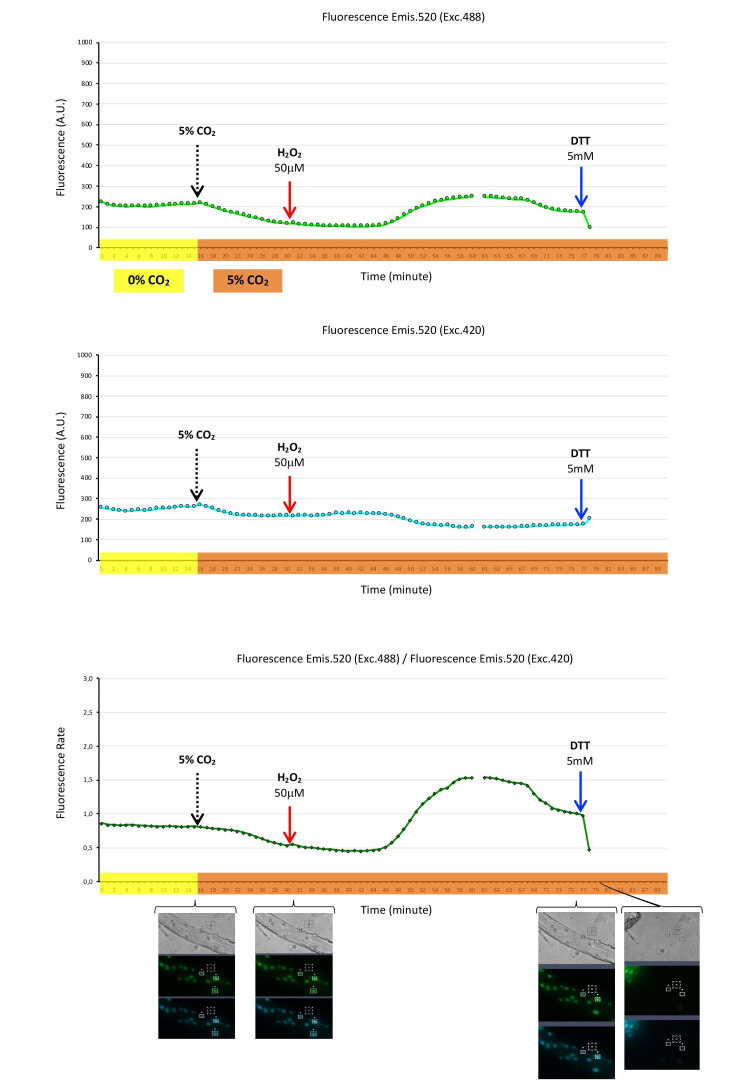
Hydrogen peroxide biosensor HyPer-nuc expressed in the nuclei of single skeletal muscle fibers. Nuclear fluorescence registered every minute during a 90 min time course, constituted by two consecutive time courses of 60 and 30 min. Fibers were maintained at 37 °C in a 0% CO_2_ atmosphere (timeline highlighted in yellow) for the first 15 min, followed by exchange to a 5% CO_2_ atmosphere (timeline highlighted in orange) for 75 min. Hydrogen peroxide (50 µM) was added to the fiber medium at the 30 min timepoint, and DTT (5 mM) at the 76 min timepoint. Top panel presents HyPer-nuc fluorescence emission at 520 nm when excited at 488 nm. Fluorescence at every timepoint is the mean of fluorescence in arbitrary units (A.U.) from three nuclei of the fiber. Middle panel presents HyPer-nuc fluorescence emission at 520 nm when excited at 420 nm. Bottom panel presents the HyPer-nuc fluorescence rate: Fluorescence emission at 520 nm when excited at 488 nm divided by fluorescence emission at 520 nm when excited at 420 nm. Sets of fiber images: Transmission (grey), green fluorescence (emis. 520, exc. 488), and blue fluorescence (emis. 520, exc. 420) at timepoints 15, 30, 76, and 79 min, where the fluorescence of the nuclei and three nuclei selected to quantify fluorescence and background fluorescence are shown. (Appendix A).

**Figure 4 ijms-22-10876-f004:**
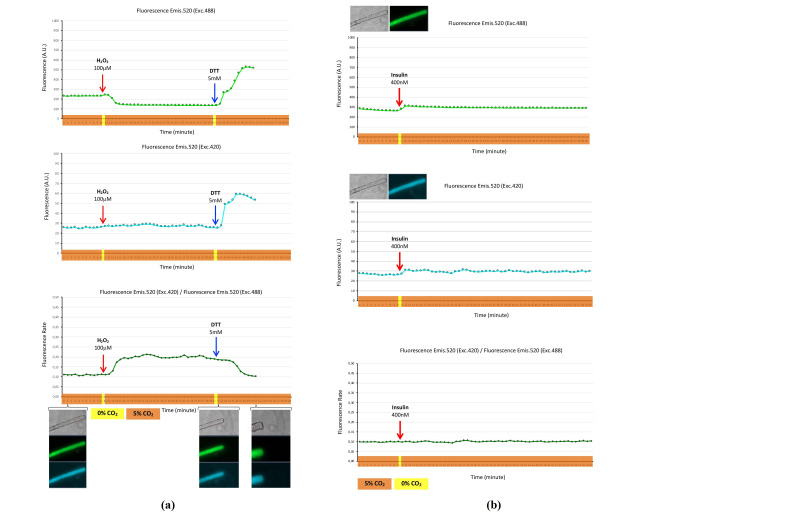
Hydrogen peroxide biosensor roGFP2-Orp1 expressed in the cytosol of single skeletal muscle fibers. Fluorescence monitoring during a time course of 60 min and fluorescence image registration every minute. Fibers were maintained at 37 °C in a 5% CO_2_ atmosphere (timeline highlighted in orange), with the exception of the timepoints in which reagents (100 µM of H_2_O_2_ or 5 mM of DTT) were incorporated into the fiber medium and the fibers were maintained at 37 °C in a 0% CO_2_ atmosphere for 40 s (timeline highlighted in yellow), the time needed to add reagents. H_2_O_2_ (100 µM final) and DTT (5 mM final) were added into the fiber medium just after the 10 and 40 min timepoints, respectively. (**a**) Top panel presents the fluorescence, in arbitrary units (A.U.), of the roGFP2-Orp1 biosensor fluorescence emission at 520 nm when excited at 488 nm of the fiber that expressed roGFP2-Orp1. Middle panel presents the fluorescence emission at 520 nm when excited at 420 nm. Bottom panel is the fluorescence rate (fluorescence emission at 520 nm when excited at 420 nm divided by fluorescence emission at 520 nm when excited at 488 nm). Sets of fiber images: Transmission (grey), green fluorescence (emis. 520, exc. 488), and blue fluorescence (emis. 520, exc. 420) at timepoints 1, 41, and 51 min, the area of the fiber selected for fluorescence quantification and the area of the background fluorescence where is shown; (**b**) top panel: Fluorescence, in arbitrary units (A.U.), of the roGFP2-Orp1 biosensor fluorescence emission at 520 nm when excited at 488 nm of the fiber that expressed roGFP2-Orp1—transmission and fluorescence images with areas selected for fluorescence quantification are shown. Similarly, the middle panel presents the fluorescence emission at 520 nm when excited at 420 nm. Bottom panel is the fluorescence rate (fluorescence emission at 520 nm when excited at 420 nm divided by fluorescence emission at 520 nm when excited at 488 nm). Insulin was added into the fiber medium (400 nM final) just after the 10 min timepoint. (Appendix A).

**Figure 5 ijms-22-10876-f005:**
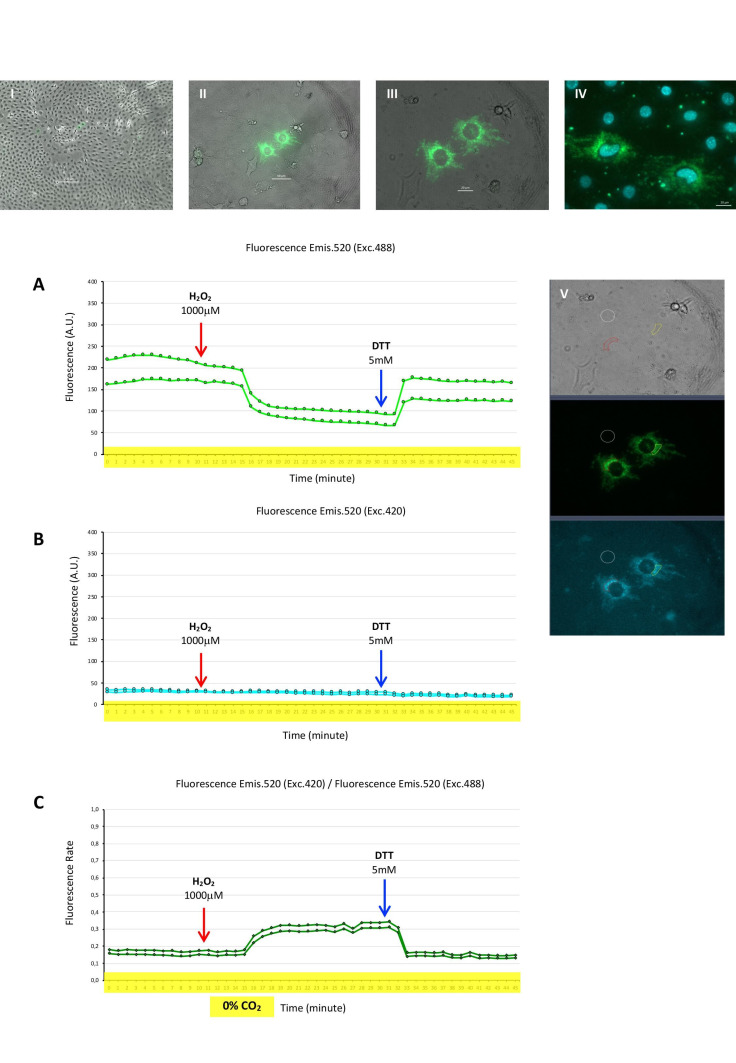
Biosensor mito-Grx1-roGFP2 for monitoring the redox potential GSH/GSSG in the mitochondria of C2C12 myoblasts. (**I**–**III**) The same two C2C12 myoblasts that expressed the biosensor mito-Grx1-roGFP2 at different magnifications. (**IV**) Two C2C12 myoblasts expressing mito-Grx1-roGFP2, stained with DAPI (blue nuclear location). Fluorescence of the mito-Grx1-roGFP2 biosensor was monitored from each of the two myoblasts that expressed mito-Grx1-roGFP2 (**I**–**III**); myoblasts were maintained at 37 °C in a 0% CO_2_ atmosphere (timeline highlighted in yellow). H_2_O_2_ (1000 µM final) and DTT (5 mM final) were added into the myoblast medium just after the 10 and 30 min timepoints, respectively. (**A**) The above panel presents the fluorescence, in arbitrary units (A.U.), of mito-Grx1-roGFP2 fluorescence emission at 520 nm when excited at 488 nm of each of the two myoblasts. (**B**) Middle panel presents the fluorescence emission at 520 nm when excited at 420 nm. (**C**) Bottom panel presents the fluorescence rate (fluorescence emission at 520 nm when excited at 420 nm divided by fluorescence emission at 520 nm when excited at 488 nm). (**V**) Set of myoblast images: Transmission (grey), green fluorescence (emis. 520, exc. 488), and blue fluorescence (emis. 520, exc. 420) show the area of mitochondria selected for fluorescence quantification and the area of background fluorescence. (Appendix A).

**Figure 6 ijms-22-10876-f006:**
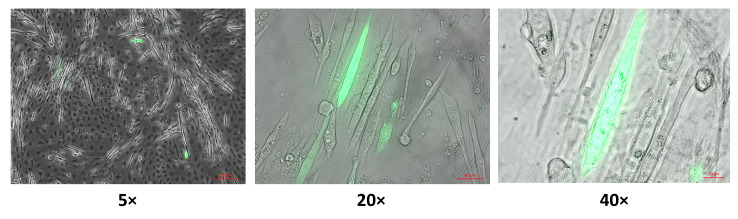
Cyto-Grx1-roGFP2 (cytosolic GSH/GSSG biosensor) expressed in C2C12 myotubes. Myotubes were obtained by experimental induction to differentiation of C2C12 myoblasts that already expressed cyto-Grx1-roGFP2, to myotubes. 5×, 20×, and 40× magnification of the C2C12 myotubes expressing cyto-Grx1-roGFP2. (Appendix A).

**Figure 7 ijms-22-10876-f007:**
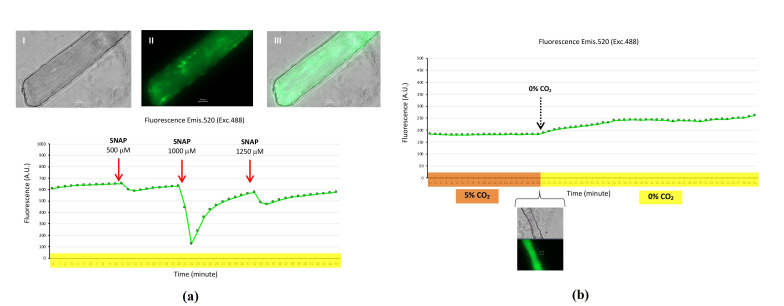
Biosensor G-geNOp for monitoring cytosolic nitric oxide in isolated skeletal muscle fibers. (**a**) A fiber that expressed G-geNOp: Transmitted light (**I**), fluorescence (**II**), and combined transmitted light and fluorescence (**III**). Graphic presents the registration of fluorescence emission at 520 nm (excitation 488 nM) in arbitrary units (A.U.) of that fiber that expressed G-geNOp during a 45 min time course where the fibers were maintained at 37 °C in a 0% CO_2_ atmosphere (timeline highlighted in yellow). The nitric oxide donor SNAP was added at different timepoints, with final concentrations of 500 µM (10 min), 1000 µM (20 min), and 1250 μM (31 min). (**b**) Monitoring fluorescence emission at 520 nm (excitation 488 nM) in arbitrary units (A.U.) from a fiber that expressed G-geNOp during a 60 min time course, with fluorescence registration every min. The first 20 min fiber was maintained at 37 °C in a 5% CO_2_ atmosphere (timeline highlighted in orange) and switched to 37 °C in a 5% CO_2_ atmosphere for the following 40 min of the time course (timeline highlighted in yellow). Fiber images: Transmitted and fluorescence, at the 20 min timepoint, where the area of the fiber selected for fluorescence quantification is shown. (Appendix A).

## Data Availability

Excluded.

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
