# Peer review of "Genetically Encoded Biosensors to Monitor Intracellular Reactive Oxygen and Nitrogen Species and Glutathione Redox Potential in Skeletal Muscle Cells"

_ijms, 2021, doi:10.3390/ijms221910876_

Round 1
Reviewer 1 Report
The subject of the article is attractive, once there is a lack of spatial and temporal information regarding the flux of nitric oxide in the cell. The data show that genetically encoded biosensors and quantitative fluorescence microscopy provide a robust methodology to investigate pathophysiological processes associated to redox biology of skeletal muscle.
General comments:
- In the Abstract is good.
- The introduction pleased explain the sentence” Regarding reactive nitrogen species (RNS), the most important are nitric oxide (NO), which is a free radical, and peroxynitrite anion (ONOO-), which it is not a free radical.”
- The main objective is unclear. The last sentence of the introduction is confusing.
- The introduction does not have the clinical advantage of this study.
- The Florescence images do not scale or are they not visible?
- In the figures the axis Time (minute) are not visible.
- Some of the images are too small, or make them larger to be visible or can be placed in supplementary material
- The clinical importance of the new methodology is never mentioned, in the skeletal muscle or in the cardiovascular system.
- The conclusion is good.
Author Response
Please, see the attachment.

Reviewer 2 Report
This is a very interesting and useful methodological work showing the promise of using a number of genetically encoded biosensors to monitor intracellular RONS and glutathione redox potential in skeletal muscle cells. The work is well written, however I have a number of questions for the methodological part and other comments.
My main criticism to the pH effect on the operation of sensors. This question occupies an essential part of the manuscript. Can the authors experimentally evaluate the effect of pH on sensor function? So far, their explanations look very speculative.
In addition, the data given in part 2.2 (HyPer-mito) are not obvious to me. I suggest that the authors take a more specific approach and induce peroxide production in mitochondria using respiratory chain inhibitors such as antimycin A.
The authors should provide experimental data showing the integrity of the cell membranes. This is important in order to exclude possible penetration of hydrogen peroxide through membrane defects.
Minor comments:
Line 56. «mitochondria appear to be the major source of ROS production»
It must be confirmed with the relevant references.
Line 57. «NADPH oxidases (NOXs), which are localized in the plasma membrane [6].»
I don't quite agree, NOXs located within the sarcoplasmic reticulum, transverse tubules, and the sarcolemma (PMID: 15780757 and PMID: 18923182). This must be taken into account.
Moreover, the authors for some reason do not mention the role of phospholipase A2 (PMID: 21167935) and xanthine oxidase in the generation of ROS (PMID: 15932896).
Line 80. "moreover many of the pathways are waiting to be discovered." This is a very loose statement and should be deleted.
Line 138. At the first mention, it is necessary to explain why the DTT was added.
Line 238. «probably due to toxicity evoked by DTT». This needs to be explained in more detail.
Author Response
Please, see the attachment.

Round 2
Reviewer 1 Report
The authors have revised the manuscript according to the reviewer's comments. However, as I mentioned in the previous review, the importance of this tool in the study of pathologies, mainly at the cardiovascular level, should be added.
Author Response
Please, see the attachment.

Reviewer 2 Report
The authors should provide data on the change in the color of the pH indicator, at least in the form of a supplementary file. I also agree with the first reviewer, the figures should be enlarged. This can also be done in the Supplementary file.
Finally, I highly recommend conducting experiments with mitochondrial superoxide generation using available inducers. Otherwise, the authors should note that the addition of exogenous hydrogen peroxide is not an adequate approach for assessing the level of ROS production by mitochondria.
Author Response
Please, see the attachment.

Round 3
Reviewer 2 Report
Overall, I am satisfied with the changes made. However, I still think that experiments with hydrogen peroxide on mitochondria are insufficient and confirmation of these data with the help of endogenous inducers of mitochondrial ROS is required. I believe that the authors will do this work in the following publications. But now they need to show the limitations of the approach used in the manuscript (pages 6-7 and discussion section) and describe the need to use endogenous ROS inductors for correct interpretation of the functioning of the proposed sensor.
The authors also forgot to submit a file with enlarged images.
Author Response
Please, see the attachment.

Round 4
Reviewer 2 Report
I have no more questions.